# Effect of Nb on the Damping Property and Pseudoelasticity of a Porous Ni-Ti Shape Memory Alloy

**DOI:** 10.3390/ma16145057

**Published:** 2023-07-17

**Authors:** Peng Sun, Qingzhou Wang, Jianhang Feng, Puguang Ji, Jianjun Zhang, Fuxing Yin

**Affiliations:** 1Tianjin Key Laboratory of Materials Laminating Fabrication and Interface Control Technology, School of Materials Science and Engineering, Hebei University of Technology, Tianjin 300130, China; sunptj@126.com (P.S.); jianhang_feng@126.com (J.F.); jipuguang@hebut.edu.cn (P.J.); zjjyyds888@163.com (J.Z.); 2Xuzhou Jihua Metal Material Technology Co., Ltd., Xuzhou 221100, China; 3Institute of New Materials, Guangdong Academy of Sciences, Guangzhou 510651, China

**Keywords:** porous shape memory alloy, Ni-Ti, powder metallurgy, damping property, pseudoelasticity

## Abstract

In order to develop novel high damping materials with excellent pseudoelasticity (PE) properties to meet the application requirements in aerospace, medical, military and other fields, porous Ni_50.8_Ti_49.2_ shape memory alloy (SMA) was prepared by the powder metallurgy method. Different contents of Nb element were added to regulate the microstructures. It was found that after adding the Nb element, the number of precipitates significantly decreased, and the Nb element was mainly distributed in the Ni-Ti matrix in the form of β-Nb blocks surrounded by Nb-rich layers. Property tests showed that with the increase in Nb content, the damping and PE increased first and then decreased. When the Nb content reached 9.0 at.%, the highest damping and the best PE could be achieved. Compared with the porous Ni-Ti SMA without Nb addition, the damping and PE increased by 60% and 35%, respectively. Correlated mechanisms were discussed.

## 1. Introduction

With the development of industry, problems such as vibration and noise are becoming increasingly serious. Damping materials can effectively convert mechanical energy into thermal energy, thus achieving the effect of vibration and noise reduction. Polymer materials exhibit excellent damping properties due to viscoelasticity, but owing to their low mechanical properties they cannot be directly used as structural components in engineering fields [1]. Although traditional metal materials have excellent mechanical properties, their applications are also limited due to their low damping properties [2,3]. SMAs exhibit high damping properties owing to their inherent thermoelastic martensitic transformation (MT) behavior and the hysteretic migration of a large number of interfaces such as the austenite/martensite interface, the twin interface as well as the martensite/martensite interface, so they have been widely used [4,5]. Compared with iron-based and copper-based SMAs, Ni-Ti SMAs with an almost equal atomic ratio have unique advantages because they have a better shape memory effect (SME) and PE besides high damping properties [6]. Moreover, the one-step reverse MT (B19′ → B2) of the Ni-Ti SMAs during the heating process can be easily changed to a two-step reverse MT (B19′ → R → B2) by performing certain heat treatments [7]. Therefore, the regulation of its damping is more convenient than other SMAs.

With the development of science and technology, aerospace, medical, military and other fields have put forward higher requirements for the damping property of metal materials [2]. It has been shown that making Ni-Ti SMAs porous can further improve their damping properties. This is because the microplastic deformation of the pore wall as well as the motion of atoms and stress mode transformation around pores will dissipate additional mechanical energy [8,9]. Element addition is another effective method of improving the damping properties of Ni-Ti SMAs. For example, the addition of Cu can significantly improve the damping of Ni-Ti SMAs, but excess Cu will embrittle the alloy and reduce the machinability [10]. The addition of Hf and H elements can form a new damping peak in Ni-Ti SMAs, but the resultant alloys have low modulus and poor deformability in the martensitic state [11]. In recent years, the addition of Nb to Ni-Ti SMAs has attracted ever more attention because of the excellent comprehensive properties of the resultant alloys. By comparing the damping properties of the Ni_50_Ti_50_ SMA and the Ni_47_Ti_44_Nb_9_ SMA, Cai et al. found that the addition of Nb contributed to an improvement in the damping property of the Ni-Ti SMA due to the formation of a high density of dislocations around the Nb phase [12]. Bao et al. fabricated a NiTiNb SMA through casting and hot rolling processes, and they found that the generation of the eutectic microstructure around the Nb phase could lead to an improvement in the damping [13].

Nevertheless, up to now, studies on the effect of Nb addition on damping properties have mostly focused on bulk Ni-Ti SMAs, and there are still few studies concerning porous Ni-Ti SMAs. In addition to damping properties, as mentioned above, the SME and PE are also important functional properties of Ni-Ti SMAs. In particular, the PE of porous Ni-Ti SMAs has shown an important application prospect in the field of high overload resistance related to high-speed launch in recent years [14]. Guo et al. demonstrated that Ni-Ti SMAs have good energy absorption ability and self-recovery capacity under dynamic loading, which can effectively improve the service life and material utilization of protective structures [15]. However, so far, there have been few studies on the PE property of porous Ni-Ti SMAs. In the present study, the effect of Nb addition on the microstructure, damping and PE properties of a porous NiTi SMA was systematically investigated. The obtained results can provide a theoretical basis for improving the comprehensive properties of the porous NiTi SMAs.

## 2. Materials and Methods

### 2.1. Fabrication of Porous Ni-Ti SMAs

Porous Ni_50.8_Ti_49.2_ SMAs with Nb addition were fabricated via the powder metallurgy process by using pure Ni, Ti and Nb powders (the purity was 99.9% and the average particle size was 75 μm) as raw materials. Powders were purchased from the Gripm Advanced Materials Co., Ltd. (Beijing, China). Ni, Ti powders and different contents (0, 3.0, 6.0, 9.0 and 12.0 at.%) of Nb powders were ball milled for 10 h on a planetary ball mill under the protection of argon (the weight ratio of ball to powder was 10:1, and the rotating speed of the ball mill was 200 rpm). Subsequently, the obtained Ni-Ti-Nb powders were homogeneously mixed with NaCl space-holders with an average particle size of 0.60 mm purchased from the China National Salt Industry Group Co., Ltd. (Beijing, China) in a V-type mixer. The mixed powders were then compressed into cylindrical green compacts under the pressure of 350 MPa on a computer numerical control (CNC) hydraulic machine purchased from the Henan Oukai Hydraulic Equipment Co., Ltd. (Xinxiang, China). The green compacts were sintered in a tube furnace purchased from the Tianjin Zhonghuan Electric Furnace Co., Ltd. (Tianjin, China) under the protection of high purity argon gas (the green compacts were firstly sintered at 790 °C for 2 h to densify the metal matrix, and then sintered at 1100 °C for 3 h). After natural cooling to room temperature, the specimens were taken out of the furnace and the residual NaCl in the pores of specimens was washed out with running water. In order to eliminate the precipitates in the Ni-Ti matrix, the porous Ni-Ti SMAs were subjected to solution treatment at 1000 °C for 60 min and then quenched in water. After that, the specimens were aged at 450 °C for 30 min. For convenience, the porous Ni-Ti SMAs with different contents (0, 3.0, 6.0, 9.0 and 12.0 at.%) of Nb were termed as Nb0, Nb3, Nb6, Nb9 and Nb12 specimens, respectively.

### 2.2. Characterization of Specimens

The porosity of porous Ni-Ti SMAs was calculated by using the following equation:(1)$$P=(1-\frac{\rho^{*}}{\rho_{0}})\times 100\%$$
where *P* and $\rho^{*}$ denote the porosity and apparent density of the porous Ni-Ti SMAs, respectively, and $\rho_{0}$ denotes the theoretical density of the dense Ni-Ti SMA.

The phase composition of specimens was analyzed by X-ray diffraction (XRD, Bruker D8 Discover). The equipment was sourced from the Bruker Corporation (Billerica, MA, USA). The pore morphology and microstructure of specimens were observed by scanning electron microscopy (SEM, JSM-7100F) equipped with an energy dispersive spectrometer (EDS). Elements mapping was obtained by using electron probe microanalysis (EPMA, JEOL 8530F). Both SEM and EPMA were sourced from the JEOL Japan Electronics Co., Ltd. (Tokyo, Japan).

### 2.3. Property Tests

The damping property was characterized by the internal friction IF (Q^−1^) and measured on a dynamic thermo mechanical analyzer (DMA, Q800) by using the method of forced vibration with a heating rate of 5°/min and a vibration frequency of 0.5 Hz. The DMA sourced from the TA Instruments (New Castle, DE, USA) is shown in Figure 1. The damping specimens had a dimension of 35 × 10 × 2 mm^3^. The PE of the specimens was tested at room temperature on a hot simulated testing machine (HSTM, Gleeble 3180) with a strain rate of 1.0 × 10^−4^ s^−1^. The equipment was sourced from the Dynamic Systems Inc. (Poughkeepsie, NY, USA). Four pre-strains of 2.0%, 3.0%, 4.0% and 5.0% were used. The cylinder specimens had a dimension of ∅8 × 12 mm^3^.

## 3. Results and Discussion

### 3.1. Phase Composition and Microstructures of Specimens

The XRD patterns of Ni-Ti and Ni-Ti-Nb (Nb content: 9.0 at.%) powders as well as porous Ni-Ti SMAs are shown in Figure 2. The sharp characteristic diffraction peaks of Ni, Ti and Nb elements indicate that the ball-milled powders have an ordered lattice arrangement. The position of the peaks without deviation indicates that the mixed powder has no interatomic diffusion behavior. In Figure 2b, diffraction peaks of Na and Cl are not detected, indicating that the NaCl particles only play the role of space-holders and do not participate in the reaction between metal powders at a high sintering temperature. The porous Ni-Ti SMAs with Nb addition are mainly composed of NiTi(B2) matrix phase and a small amount of Ti-rich phase and β-Nb phase. The Ti-rich phase is probably (Ti, Nb)_2_Ni phase [16]. The existence of β-Nb phase indicates that the sintering temperature of 1100 °C could not provide enough energy for the Nb element to form an eutectic structure with the NiTi phase. Compared with the specimens with the Nb addition, a higher content of the Ni-rich phase appears in the Nb0 specimen. For the specimens with Ni content higher than 50.5 at.%, the solution treatment can make part of the Ni_3_Ti phase re-dissolve in the NiTi phase [17,18]. However, in the subsequent aging treatment at 450 °C, the Ni-rich phase precipitates again as the Ni_4_Ti_3_ phase [19]. Interestingly, in this study the specimens with the Nb addition do not show the diffraction peaks of Ni_3_Ti and Ni_4_Ti_3_ phases after heat treatment, indicating that the presence of the Nb element in the matrix can effectively hinder the precipitation of Ni-rich phases.

Figure 3a,b shows the morphologies of Ni-Ti and Ni-Ti-Nb (Nb content: 9at.%) powders, respectively. The two kinds of powders are irregularly shaped and have no obvious difference in the morphology. During the compressing process, the irregular shape gives a better meshing effect between the powders, which can effectively improve the strength and density of the compacts and can avoid the delamination and cracking of the compacts during the demolding process. Figure 3c shows the morphology of NaCl particles with an average particle size of 0.6 mm. Figure 3d–g shows the macro and micro morphologies of the Nb0 specimen, and Figure 3h–k shows the macro and micro morphologies of the Nb3 specimen. There is no significant difference in the macroscopic morphology between the two specimens. The pores are uniformly distributed in the Ni-Ti matrix and connected with each other, forming a three-dimensional network. The pores of the specimens retain the regular three-dimensional shape of the NaCl particles while losing their sharp-edged characteristics. This may be due to the fragmentation of the edges of the NaCl particles during the compressing process. The inner wall of pores is very rough, which is conducive to improving the ability of the material to absorb sound and reduce noise [20]. A 1100 °C sintering temperature provides enough energy for the formation and growth of the sintered neck. The powder particles exhibit a dense metallic framework due to good metallurgical bonding.

When we observe the micro-morphology of the pore wall at a higher magnification, it is clear that the Nb3 specimen has fewer micro pores than the Nb0 specimen, which can be attributed to the fact that the presence of Nb promotes the diffusion of atoms between Ni and Ti; this effectively reduces the vacancies generated by the Kirkendall effect during the high temperature sintering process [21]. Although Nb has a higher melting point than Ni and Ti, Nb and Ti belong to an infinite solid solution system, so Nb may still be dissolved in the Ni-Ti matrix [22]. Nb facilitates the sintering process by acting as a kind of channel for the diffusion and reaction between Ni and Ti. Thus, the migration of atoms in the powders is accelerated, which causes the distance between the powders of different elements to decrease continuously, resulting in the gradual disappearance of micro pores.

Figure 4 shows the SEM images (backscattered electron images) of the porous Ni-Ti SMAs with different Nb contents. It can be seen from Figure 4a,b that the Nb0 specimen is mainly composed of a gray matrix and a small number of white precipitations. Combining the XRD analysis results shown in Figure 2b and the EDS analysis results of points A–D shown in Figure 5a, it can be determined that the gray matrix is the NiTi phase, whereas the white Ti-rich phase at point B, the flaky Ni-rich phase at point C, and the fine acicular Ni-rich phase at point D are the Ti_2_Ni, Ni_3_Ti and Ni_4_Ti_3_ phases, respectively [23,24]. From Figure 4b, it can be seen that the size of the Ni_4_Ti_3_ phase is 2–3 μm.

Figure 4c–g shows the backscattered electron images of the Nb3, Nb6, Nb9 and Nb12 specimens, respectively. In these specimens, white blocks appear, and with the increase in Nb content, the number of these white blocks increases. Figure 4d shows the image of the Nb3 specimen with higher magnification, and Figure 5b gives the EDS analysis results of points E–H. According to Figure 2b and the EDS result of point E, it can be determined that these white blocks are the β-Nb phase. In addition to the blocky β-Nb phase, a striped β-Nb phase is also found in Figure 4f,g, which may be related to the creep of the β-Nb phase during the sintering process [16]. From Figure 4d, it is apparent that the β-Nb phase is surrounded by a layer of Nb-rich phase. The clearly discernible transition region (the Nb-rich phase) increases the bonding effect between the β-Nb phase and the NiTi matrix. Moreover, Nb can also dissolve in NiTi and Ti_2_Ni phases by replacing Ti and form (Ti, Nb)Ni and (Ti, Nb)_2_Ni phases, respectively. Because the sintering temperature (1100 °C) of porous Ni-Ti SMAs in this study is lower than the eutectic temperature (1150.7 °C) of the Ni-Ti-Nb alloy, the eutectic structure reported by Bao et al. is not found in Figure 4 [13,25,26].

Figure 6 shows the EPMA analysis results of the Nb0 specimen. Except for Ni and Ti, no other elements can be detected. Combining the EDS results shown in Figure 5a, it can be determined that the Ni-rich phase in Figure 6c is the Ni_3_Ti phase, and the uniformly distributed Ti-rich phase in Figure 4d is the Ti_2_Ni phase. The needle-like Ni_4_Ti_3_ phase, however, is not found in Figure 6 due to its small size. Figure 7 shows the EPMA analysis results of the Nb9 specimen. No other elements and Ni-rich phases can be detected except for the Ni, Ti, Nb elements and (Ti, Nb) Ni phase, which is in good accordance with the results shown in Figure 2b and Figure 4c–g. The Ti-rich phase shown in Figure 4d cannot be detected due to its low content and the lower magnification of Figure 5 compared with that of Figure 4d. In Figure 7d, the layer of the Nb-rich phase shown in Figure 4d between the β-Nb phase and NiTi phase can still be clearly seen.

### 3.2. Effect of Nb on the Damping Property

Figure 8 shows the IF-temperature spectrum of the porous Ni-Ti SMAs with different contents of Nb. It can be seen that during the heating process two IF peaks arise at around −40 °C and 30 °C, respectively, which can be ascribed to the two steps transformation from the B19′ phase to the B2 phase (B19′ → R → B2) [7,27]. It was reported that the Ni_4_Ti_3_ phase precipitated after aging treatment could cause a stress field in the NiTi matrix and facilitate the transformation of the NiTi phase into the R phase; the two-step phase transformation of B19′ → R → B2 therefore occurs in the Nb0 specimen [7,28]. In the specimens containing Nb, however, the precipitation of the Ni_4_Ti_3_ phase is inhibited, so only the transformation of B19′ → B2 occurs (only one IF peak appears).

From Figure 8, it is apparent that with the increase in Nb content, the damping (the level of the IF curve) of the porous Ni-Ti SMAs increases first and then decreases. When the Nb content reaches 9.0 at.%, the highest damping can be achieved. The damping capacity of the porous Ni-Ti SMAs has the following three influencing factors. One is the pore structure. Due to the existence of pores in the Ni-Ti matrix, the external stresses applied during IF measurements are not uniformly distributed in the specimens, making the strain significantly lag behind the stress and leading to the enhancement of the IF. In addition, uniformly distributed stresses will cause the pores to undergo the deformations of expansion and contraction, which are also conducive to the conversion of external mechanical energy into internal energy. However, in this study, all specimens have the same porosity. Therefore, the IF that originates from the pore structure is not the main influencing factor on the change in the IF with the Nb content. The other influencing factors are the damping sources in the matrix of the porous Ni-Ti SMAs. The hysteretic migration of interfaces in the Ni-Ti matrix is the main energy dissipation mechanism that leads to strain lagging behind stress and improvement in the damping [6]. Compared with the specimens containing the Nb element, the Nb0 specimen has a higher content of Ti_2_Ni and Ni_3_Ti phases without mobile interfaces, so the Nb0 specimen has lower damping. After adding the Nb element, as shown in Figure 4, the content of Ti_2_Ni and Ni_3_Ti phases significantly decreases. As a consequence, the damping increases. Moreover, from Figure 4d it is apparent that after adding the Nb element, a layer of Nb-rich phase appears around the β-Nb phase, forming a new interface of a β-Nb/Nb-rich layer/Ni-Ti matrix. This kind of interface is also mobile under applied stresses, so it acts as a new damping source of the porous Ni-Ti SMA [13]. It has been reported that there is a high density of dislocations in the regions adjacent to the β-Nb/Ni-Ti interface [12]. The slip of dislocations is also conductive to the improvement in the damping. Therefore, the addition of the Nb element not only decreases the content of precipitates, but also generates multiple damping sources by forming the β-Nb phase in the Ni-Ti matrix, and leads to the improvement in the damping. The third influencing factor of the damping is the existing form of the Nb element in the Ni-Ti matrix. In addition to the form of the β-Nb phase, a small number of Nb atoms dissolves into the NiTi phase. It has been shown that the dissolved Nb atoms can induce the movement of twins and effectively improve the intrinsic damping of martensite [13]. However, as the number of dissolved Nb atoms continues to increase, the damping of martensite will no longer increase, but will remain at a stable level. Therefore, dissolved Nb atoms are not the main reason for the change in damping with the Nb content. In addition, from Figure 8 it can also be seen that with the increase in the Nb content, the P_3_ peak arising from the reverse MT (B19′ → B2) gradually shifts to a low temperature side. This can be attributed to the inhibiting effect of Nb on the precipitation of Ni-rich phases. The increase in the Ni/Ti atomic ratio leads to the decreased temperature of the reverse MT [29].

The porous Ni-Ti SMAs with Nb addition can be regarded as a two-phase structure consisting of the β-Nb phase and NiTi phase. The damping arising from the β-Nb phase consists of the intrinsic damping of the β-Nb phase and the damping of the β-Nb/Nb-rich layer interface and the Nb-rich layer/Ni-Ti interface. The intrinsic damping of the β-Nb phase is very low [21], so the damping of the β-Nb/Nb-rich layer interface and the Nb-rich layer/Ni-Ti interface is the main damping source. In addition, the Nb-rich layer/Ni-Ti interface provides nucleation sites for the formation of martensite during cooling, thus creating more interfaces, which further promote the improvement of the damping. With the increase in the Nb content, the volume fraction of the β-Nb phase increases, whereas the volume fraction of the NiTi phase decreases. According to Figure 8, the damping increases first and then decreases with the increase in the Nb content. When the Nb content reaches 9.0 at.%, the highest damping is obtained. When the increase in the interface damping can compensate for the decrease in the damping caused by the decrease in the volume fraction of the NiTi phase, the damping increases, whereas when the Nb content reaches 12.0 at.%, the level of the IF curve decreases, indicating that the increase in interface damping is no longer able to compensate for the decrease in damping caused by the decrease in the volume fraction of the NiTi phase.

### 3.3. Effect of Nb on the PE Property

The PE of the Ni-Ti matrix determines the shape recovery process of the porous Ni-Ti SMAs after stress unloading, whereas the pores can be regarded as the phase that does not reflect shape recovery. Figure 9 shows the effect of Nb on the shape recovery of the porous Ni-Ti SMAs at different pre-strains (2.0%, 3.0%, 4.0% and 5.0%). As can be seen from the figure, the strain of the compressed specimen after unloading is composed of residual strain (ε_R_), superelastic reversion strain (ε_SE_), and pure elastic reversion strain (ε_E_). The reason for the high PE response of the specimens is the result of stress-induced MT and subsequent reverse MT, which is clearly reflected in the stress-strain curves during the loading–unloading processes. Taking the Nb9 specimen as an example, during the loading process, an inflection point appears on the stress-strain curve when the stress reaches 90 MPa, indicating that the stress-induced MT begins to occur. When the compressive strain reaches 5%, the stress is unloaded. Subsequently, the reverse MT begins to occur, and the strain decreases linearly to about 3.36%. After that, the strain continues to recover until reaching the ε_R_ of 1.77%. Then ε_E_, ε_SE_ and the total shape reversion strain can be determined as 1.64%, 1.59% and 3.23%, respectively, so the total recovery rate can reach 64.6% when the pre-strain is 5%.

The superelastic recovery strain of different specimens after unloading under the four test conditions is summarized in Figure 10. It can be seen that with the increase in the Nb content, the superelastic recovery strain increases first and then decreases. The Nb9 specimen has the highest superelastic recovery strain, indicating that Nb has a positive contribution to the PE. In addition, the Ti_2_Ni and Ni_3_Ti phases in the Nb0 specimen have no PE, so the reduction in the number of these phases also contributes to the improvement in the PE.

It has been reported that in Ni-Ti SMAs with the Nb addition, there is a large number of twinned martensites at the interface of the Nb-rich phase/Ni-Ti matrix [29], and this structure is considered necessary for the self-accommodation of martensite variants [30]. During loading and unloading processes, with the increase in the strain, reorientation occurs between martensite variants when the stress exceeds the yield stress of martensites. The reorientation and de-twinning of twinned martensites facilitate the hyperelastic recovery of the matrix [31]. However, the β-Nb phase has no PE, and its plastic deformation is not conducive to shape recovery. Therefore, as the content of the β-Nb phase continues to increase and the volume fraction of the Ni-Ti matrix phase continues to decrease, the PE of the porous Ni-Ti SMAs will decrease. Moreover, from Figure 9 it can be seen that the level of stress-strain curves increases first and then decreases with the increase in the Nb content. The increase in stress-strain curves can be attributed to the solid solution strengthening effect of Nb atoms as well as the dislocation strengthening effect due to the thermal mismatch between the β-Nb phase and the Ni-Ti matrix [29]. However, an excessive soft β-Nb phase will inevitably degrade the strength of the porous Ni-Ti SMAs. Therefore, the Nb9 sample has the highest level of stress-strain curve. Clearly, the strengthening of the porous Ni-Ti alloy contributes to the improvement in PE.

## 4. Conclusions

Porous Ni-Ti SMAs with different Nb contents were fabricated. The pores were uniformly distributed in the Ni-Ti matrix and connected with each other, forming a three-dimensional network. The addition of Nb can effectively reduce the numbers of micro pores and precipitates in the Ni-Ti matrix.With the increase in Nb content, the IF peak of the porous Ni-Ti SMAs arising from the reverse MT gradually shifts towards the low temperature side. This can be attributed to the hindering effect of Nb on the precipitation of Ni-rich phases as well as the replacement of Nb atoms on Ti atoms.With the increase in Nb content, both the damping and the PE of the porous Ni-Ti SMAs increase first and then decrease. The Nb9 specimen has the highest damping and the best PE. The improvement in damping can be ascribed to the formation of mobile interfaces around β-Nb phase, whereas the improvement in PE is related to the formation of twinned martensites and the strengthening of the NiTi matrix. In addition, the decrease in the content of low damping Ni-rich phases without PE is also conductive to the improvements in damping and PE. The decrease in damping and PE is caused by the decrease in the NiTi content.

## Figures and Tables

**Figure 1 materials-16-05057-f001:**
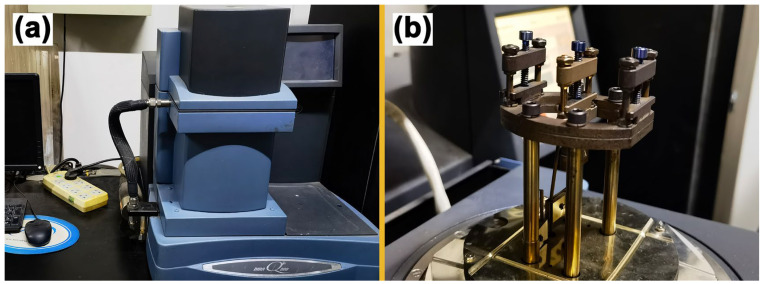
Images of (**a**) DMA, (**b**) sample fixture on the DMA.

**Figure 2 materials-16-05057-f002:**
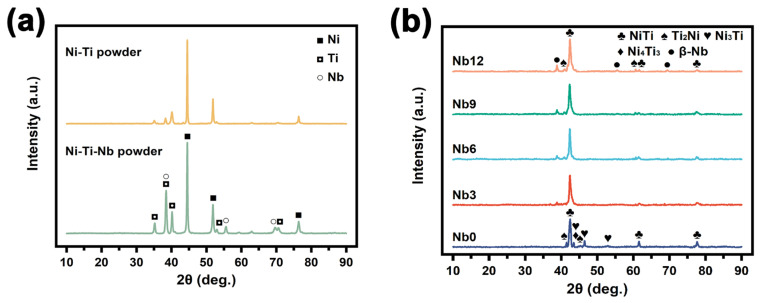
XRD patterns of (**a**) Ni-Ti and Ni-Ti-Nb powders, (**b**) porous Ni-Ti SMAs with Nb addition.

**Figure 3 materials-16-05057-f003:**
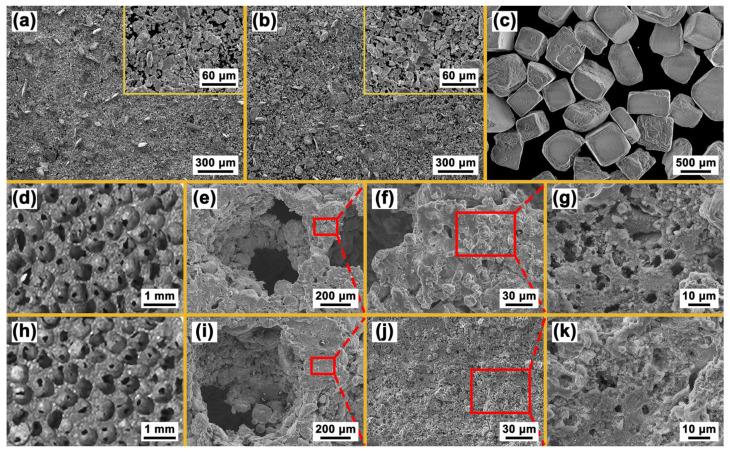
Morphologies of (**a**) Ni-Ti powders, (**b**) Ni-Ti-Nb powders, (**c**) NaCl particles, (**d**) Nb0 specimen, (**e**) pores in Nb0 specimen, (**f**) pore wall of Nb0 specimen, (**g**) micro pores in Nb0 specimen, (**h**) Nb3 specimen, (**i**) pores in Nb3 specimen, (**j**) pore wall of Nb3 specimen, (**k**) micro pores in Nb3 specimen.

**Figure 4 materials-16-05057-f004:**
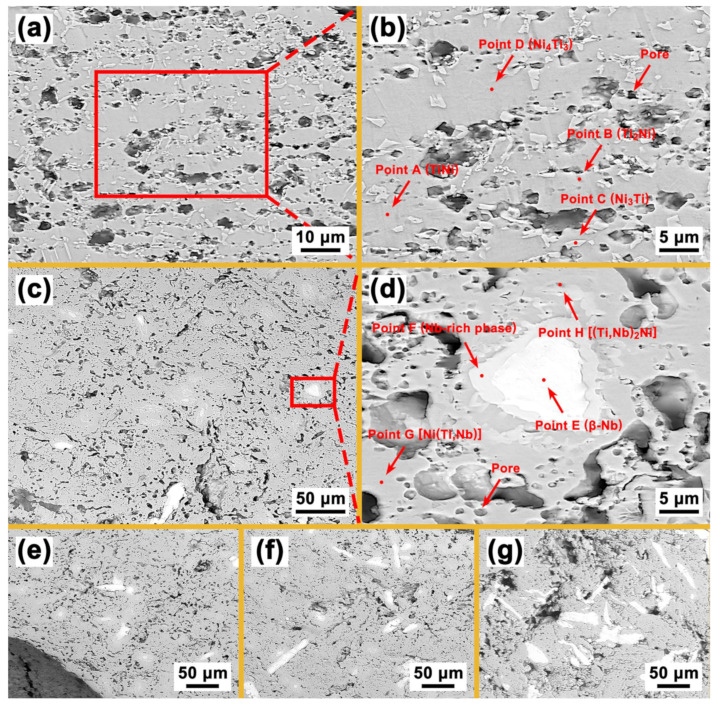
(**a**) Microstructure of Nb0 specimen; (**b**) precipitates in Nb0 specimen; (**c**) microstructure of Nb3 specimen; (**d**) precipitates in Nb3 specimen; (**e**) microstructure of Nb6 specimen; (**f**) microstructure of Nb9 specimen; (**g**) microstructure of Nb12 specimen.

**Figure 5 materials-16-05057-f005:**
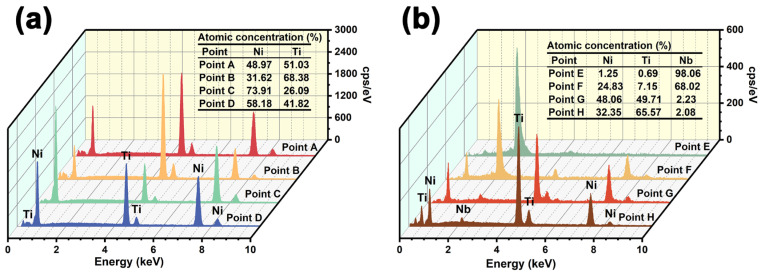
(**a**) EDS results of Nb0 specimen; (**b**) EDS results of Nb3 specimen.

**Figure 6 materials-16-05057-f006:**
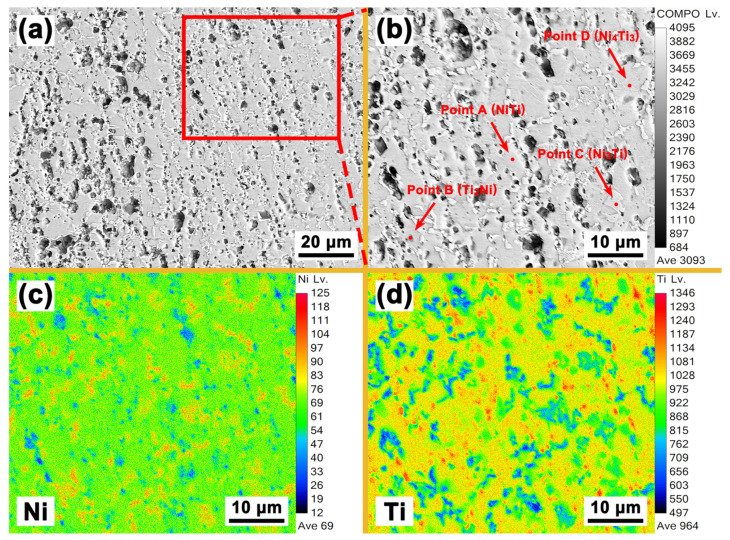
EPMA analysis results of Nb0 specimen: (**a**) microstructure; (**b**) high magnification image of the marked area in (**a**); (**c**) distribution of Ni element; (**d**) distribution of Ti element.

**Figure 7 materials-16-05057-f007:**
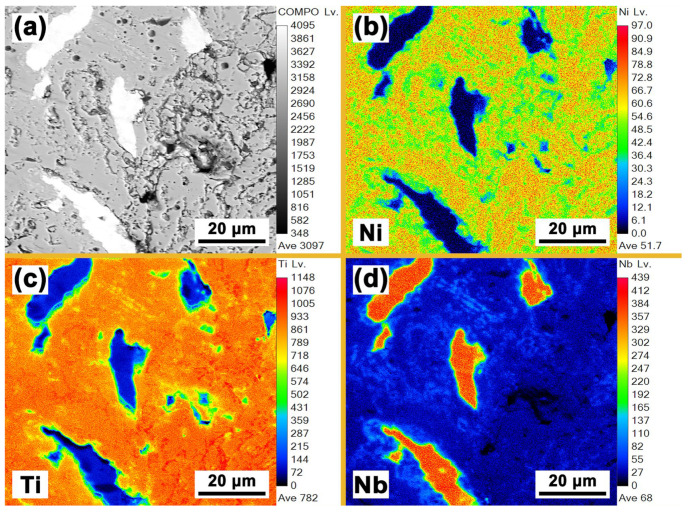
EPMA analysis results of Nb9 specimen: (**a**) microstructure; (**b**) distribution of Ni element; (**c**) distribution of Ti element; (**d**) distribution of Nb element.

**Figure 8 materials-16-05057-f008:**
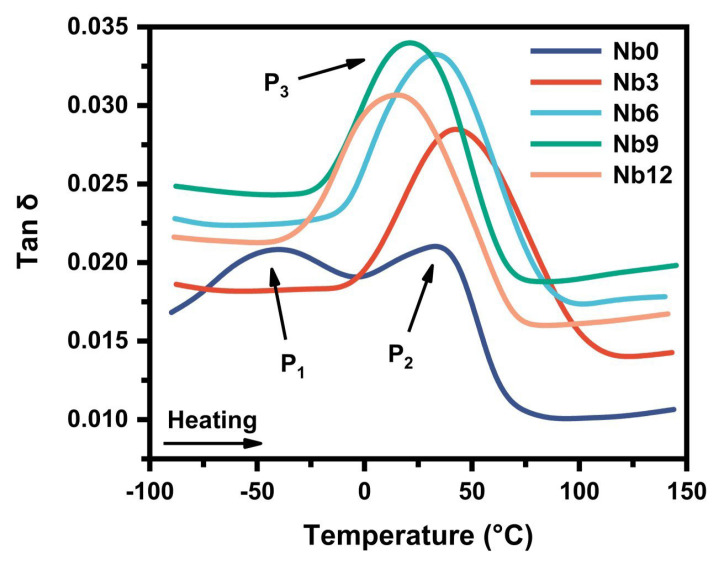
Effect of Nb on the damping property of the porous Ni-Ti SMAs.

**Figure 9 materials-16-05057-f009:**
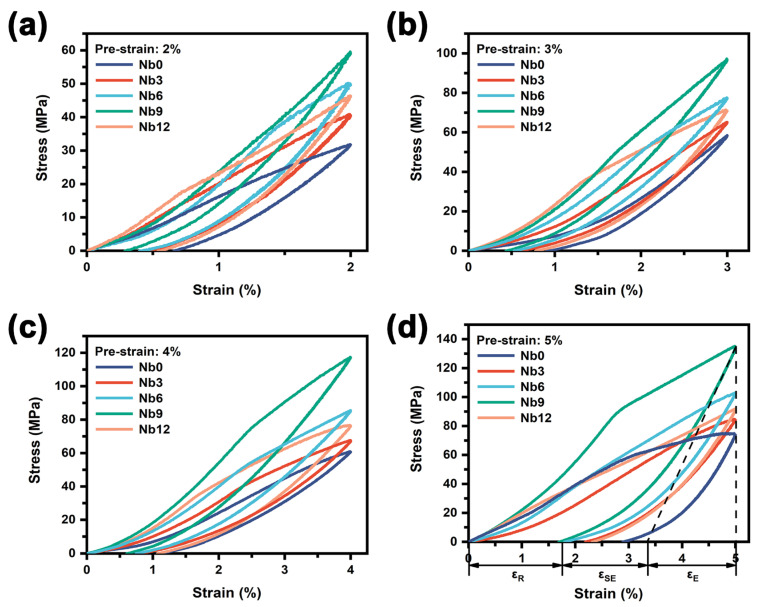
Effect of Nb on the shape recovery of the porous Ni-Ti SMAs at different pre-strains: (**a**) 2%; (**b**) 3%; (**c**) 4%; (**d**) 5%.

**Figure 10 materials-16-05057-f010:**
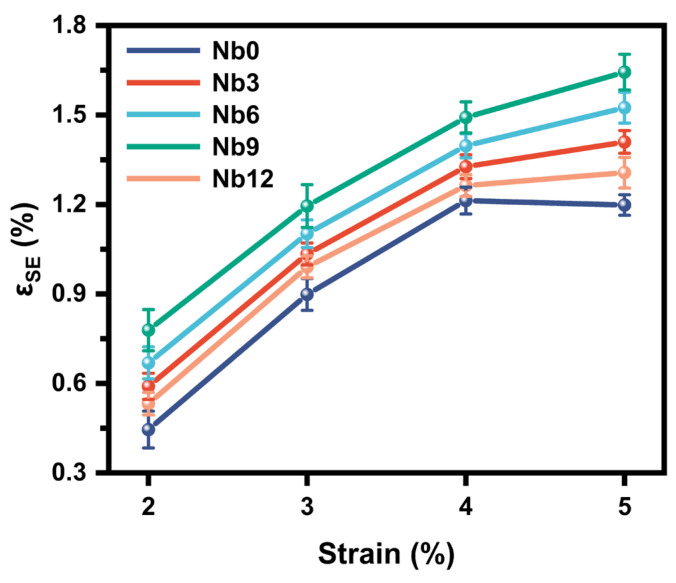
Effect of Nb content on the PE of the porous Ni-Ti SMAs.

## Data Availability

Not applicable.

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
