# Peer review of "Effect of Nb on the Damping Property and Pseudoelasticity of a Porous Ni-Ti Shape Memory Alloy"

_materials, 2023, doi:10.3390/ma16145057_

Round 1

Reviewer 1 Report

The article presents an interesting topic related to effect of Nb addition on the damping property and pseudoelasticity of a porous NiTi SMA. The manuscript is well organized, the studies are well planned, and the results are clearly reported. However, some corrections could be done to improve the paper. Some of my comments and questions on this manuscript are as follows:

2. Materials and Methods

2.1.

Line 71 – How optimal milling conditions, such as time, the weight ratio of ball to powder, and the rotating speed of ball mill, were selected?

Line 78 – How were the sintering and heat treatment parameters selected?

3. Results and Discussion

3.1.

Line 119 – Authors mentioned that „Compared with the specimens with Nb addition, higher content of Ni-rich phase appears in the Nb0 specimen”. Was a quantitative analysis of the content of individual phases in the samples performed?

Figure 3 c) and f) – EDS spectrum - Were trace amounts of oxygen identified??

Line 195 – What is the size of the Ni4Ti3 phase?

3.2.

Line 209 – Information about a two-steps martensitic transformation is mentioned for the first time in the text. Maybe it is worth mentioning briefly in the Introduction part about the one- and two-step MT and what is the R phase?

Reviewer 2 Report

1. Introduction can be improved by comparing the results of the present study with the results of previously published articles. So readers can easıly understand the results.

2.  Figure 4 seems smoothed too much, it is recommend to use the data as it is without smoothing it.

3. Remove the values of intensity from y axis, as it is just arbitrary values.

4. There are few unindexed peaks in XRD. Are they impurity peaks and authors must index them for confirmation of unspecified phases. 

5. Quality of SEM images in figure 8 is not good, recommending to go for higher maginification for the clear details about the particles.

6. In figure 9, use the r2 value as 0.999 but not as 0,999. Maintain the same throughout the figure 9 especially in the axes.

7. Conclusion must be revised by adding the important results and the novelty of the research work.

Minor

Reviewer 3 Report

·       English language should be revised. Hence there, some grammatical errors are found.

·       The abstract should be enhanced because it seems too short. Please add more informative, specific, and concise data. Also provides more context for the reader and helps them better understand the work's significance.

·       The work objective and novelty are not clear in the introduction. , the authors should illustrate this issue.

·       Figure 3 can be divided into two figures because the font in the current figure is not clear enough for readers.

·       Damping properties should be expressed in terms of zeta or correlated relationship, but in the current manuscript, the authors depend on the value that appeared from the dynamic analyzer, also tan delta, which is the main indicator for damping level; the authors don't mention from where these values are calculated.

·       The samples dimension are not given for the dynamic test, which is the standard the authors used.  

·       The dynamic experiments should be illustrated using schematic drawings or real images of the test.

·       Please explain why the damping is decreased as the temperature increases because it is not logical; hence the authors related this due to “the increase of interface damping is no longer able to compensate for the 270 decrease of damping caused by the decrease of the volume fraction of NiTi phase”.

·       Figure 7 (b) missing y-axis title

I have carefully reviewed the manuscript and have identified a few additional grammatical errors that I would like to correct before the manuscript is resubmitted.

Reviewer 4 Report

The paper titled “Effect of Nb on the Damping Property and Pseudoelasticity of a Porous Ni-Ti Shape Memory Alloy” reports new experimental results of the study aimed to improve the comprehensive properties of porous Ni-Ti shape memory alloys. I recommend it for publication after minor revision. The comments are listed below.

1.     All the abbreviations should be defined when first used (for instance, CNC hydraulic machine).

2.     For all commercial material and devices used, the information of "manufacture, city, abbreviated state (for USA/Canada), country" should be added.

3.     Figure 1b is not informative enough. It should be replotted as 2D plot, and all the reflections should be labeled.

4.     Title of the Y-axis in Figure 7b is missing. Also, I recommend to indicate the pre-strains in all plots of Figure 7.

Round 2

Reviewer 2 Report

Article seems to be in good shape

Reviewer 3 Report

the manuscript is accepted in the current from